# A Variational Perspective on Generative Protein Fitness Optimization

**Lea Bogensperger** [1]   **Dominik Narnhofer** [2]   **Ahmed Allam** [1]   **Konrad Schindler** [2]   **Michael Krauthammer** [1]

## Abstract

The goal of protein fitness optimization is to discover new protein variants with enhanced fitness for a given use. The vast search space and the sparsely populated fitness landscape, along with the discrete nature of protein sequences, pose significant challenges when trying to determine the gradient towards configurations with higher fitness. We introduce *Variational Latent Generative Protein Optimization* (VLGPO), a variational perspective on fitness optimization. Our method embeds protein sequences in a continuous latent space to enable efficient sampling from the fitness distribution and combines a (learned) flow matching prior over sequence mutations with a fitness predictor to guide optimization towards sequences with high fitness. VLGPO achieves state-of-the-art results on two different protein benchmarks of varying complexity. Moreover, the variational design with explicit prior and likelihood functions offers a flexible plug-and-play framework that can be easily customized to suit various protein design tasks.

## 1. Introduction

Protein fitness optimization seeks to improve the functionality of a protein by altering its amino acid sequence, to achieve a desired biological property of interest called "fitness" – for instance, stability, binding affinity, or catalytic efficiency. It requires searching through a vast combinatorial space (referred to as the "fitness landscape"), where the number of possible sequences grows exponentially with the sequence length $d$, while only a small subset of these sequences exhibit meaningful biological functionality (Hermes et al., 1990). Traditionally, protein fitness optimization has been addressed through directed evolution (Romero & Arnold, 2009), mimicking natural evolution in the labora-

tory with a time-consuming, yet narrow random exploration of the fitness landscape. This highlights the need for efficient in-silico methods capable of exploring the space of potential sequences, so as to prioritize promising candidates for experimental validation.

Many different approaches exist, from generative models (Jain et al., 2022; Gruver et al., 2024; Notin et al., 2022), evolutionary greedy algorithms (Sinai et al., 2020) to gradient-based sampling strategies, such as Gibbs With Gradients (GWG) (Grathwohl et al., 2021; Emami et al., 2023) and smoothed GWG variants (Kirjner et al., 2023). In general, these methods are challenged by the high-dimensional, sparse and discrete nature of the fitness landscape, characterized by ruggedness (Van Cleve & Weissman, 2015) due to epistasis as well as holes due to proteins with very low fitness (Johnston et al., 2023; Sinai & Kelsic, 2020). To deal with the inherently discrete nature of amino acids, computational methods tend to rely on discretized processes, and thus must cope with issues such as non-smooth gradients and the vast diversity of possible sequences (Kirjner et al., 2023; Frey et al., 2023).

Here we look at the problem from a different perspective: instead of relying on a token-based sequence representation, we embed sequences in a continuous latent space and learn the corresponding latent distribution in a generative manner. In this way, patterns and relationships can be captured that are difficult to model in the original discrete space, especially if one only has access to a small data set containing only a few thousand mutations of a protein.

To construct a prior distribution over latent protein sequences, we leverage *flow matching* (Lipman et al., 2023; Liu et al., 2022), a powerful generative modeling scheme that learns smooth, continuous representations amenable to gradient-based sampling and optimization. The prior is then integrated into a variational framework, making it possible to guide the search towards regions of high fitness with a fitness predictor in the form of a neural network, trained on a limited set of sequence mutations with associated fitness labels. See Figure 1. The versatility of the variational framework means that it can easily be tuned to different protein optimization tasks by suitably adapting the prior and the guidance function. We validate VLGPO on two public benchmarks for protein fitness optimization in limited data

---

[1]University of Zurich [2]ETH Zurich. Correspondence to: Lea Bogensperger <lea.bogensperger@uzh.ch>.

*Proceedings of the 42$^{nd}$ International Conference on Machine Learning*, Vancouver, Canada. PMLR 267, 2025. Copyright 2025 by the author(s).

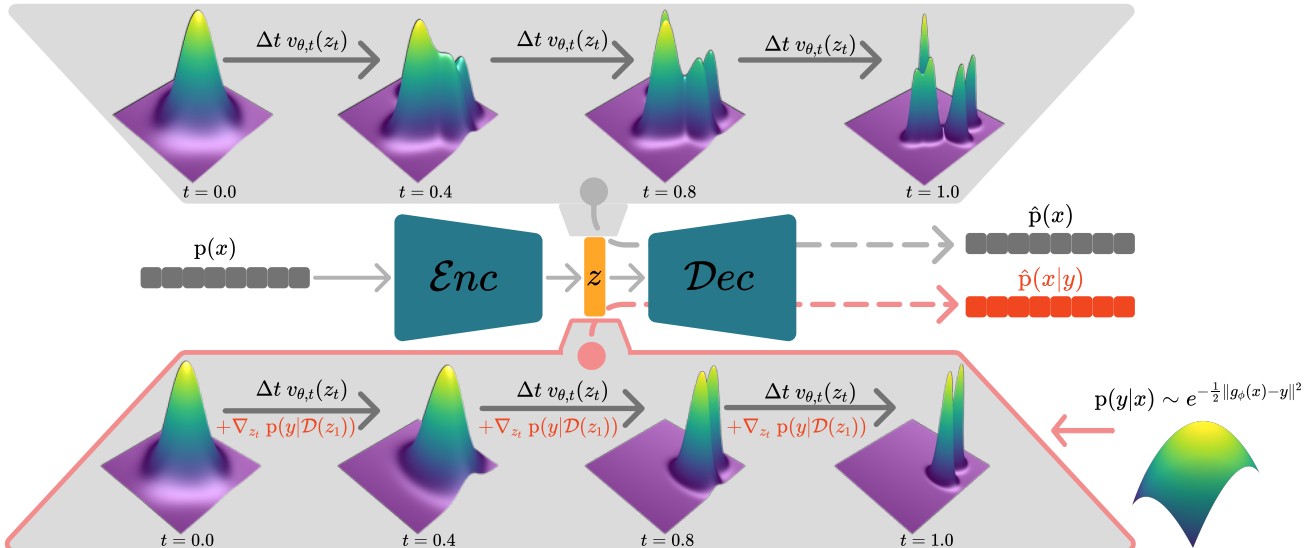

*Figure 1.* Overview of VLGPO sampling. The central section illustrates the VAE framework, showcasing protein sequences, their latent representations $z$, and the approximate posterior distribution. While the upper section depicts unconditional sampling from the prior $p(x)$ using flow matching in the latent space, the lower section illustrates the modifications introduced by VLGPO during sampling. We additionally incorporate a likelihood term $p(y|x)$ to condition on the fitness $y$, enabling sequence generation from the posterior distribution $p(x|y)$ and facilitating sampling from high-fitness regions (as shown by the shifted and reshaped distribution).

regimes, namely Adeno-Associated Virus (AAV) (Bryant et al., 2021) and Green Fluorescent Protein (GFP) (Sarkisyan et al., 2016), as suggested by (Kirjner et al., 2023).

To summarize, our contributions are:

- We introduce Variational Latent Generative Protein Optimization (VLGPO), a variational framework for protein fitness optimization that enables posterior sampling of protein sequences conditioned on desired fitness levels. Our method combines a learned prior and a fitness predictor to effectively guide the optimization towards high-fitness regions.
- We perform fitness optimization in a continuous latent representation that embeds meaningful relations between discrete protein sequences. Within this latent space, we employ flow matching to learn a generative prior that enables efficient exploration of the fitness landscape and thus facilitates the discovery of high-fitness sequences.
- We demonstrate state-of-the-art performance on established benchmarks for protein fitness optimization, namely AAV and GFP, targeting in particular tasks of medium and high difficulty in a limited data regime[1].

---

[1]Source code available at
`https://github.com/uzh-dqbm-cmi/VLGPO`.

## 2. Related Work

We consider in-silico methodologies for protein fitness optimization at the sequence level, while emphasizing that there is an extensive stream of research centered around active learning frameworks, which iteratively integrate computational predictions with experimental validations (Yang et al., 2025; Lee et al., 2024). While these methods are often considered the gold standard due to their integration of experimental feedback, our contribution is confined to purely computational strategies, thereby complementing existing active learning approaches.

In a typical directed evolution setup (Romero & Arnold, 2009), biologists simulate nature in the laboratory by running multiple rounds of mutagenesis, searching through the local landscape by sampling a few mutations away from the current position in the sequence, effectively discovering new sequences through random walks in the sequence space. This process can be resource-intensive, as many mutations do not enhance the fitness of the starting sequence, and the vast number of combinations makes exhaustive exploration impractical. Therefore, many techniques have been introduced to address these challenges by employing surrogate models to guide the search more efficiently (Sinai et al., 2020; Brookes et al., 2019; Trabucco et al., 2021; Ren et al., 2022). For instance, Adalead (Sinai et al., 2020) uses a black-box predictor model to inform a greedy algorithm to prioritize mutations that are more likely to improve protein fitness.

Moreover, generative methods have been explored for sequence generation. For instance, GFlowNets (GFN-AL) (Jain et al., 2022) are designed to sample discrete objects, such as biological sequences, with probabilities proportional to a given reward function, facilitating the generation of diverse and high-quality candidates. Further, discrete diffusion models have been used for sequence sampling, where gradients in the hidden states of the denoising network help to guide the sequence design (Gruver et al., 2024). Alternatively, a walk-jump algorithm was proposed to learn the distribution of one-hot encoded sequences using a single noise level and a Tweedie step to recover the sampled sequences after MCMC sampling (Frey et al., 2023), which was further extended using gradient guidance in the noisy manifold (Ikram et al., 2024). Likewise, also GWG proposes a strategy to obtain gradients for MCMC sampling of discrete sequences (Grathwohl et al., 2021; Emami et al., 2023). The method Gibbs sampling with Graph-based Smoothing (GGS) builds upon this by additionally regularizing the noisy fitness landscape with graph-based smoothing (Kirjner et al., 2023).

Since the search space of protein sequences grows with sequence length, many works have started to recognize the significance of a latent space (Praljak et al., 2023) or embedding spaces that are used in large-scale protein language models (Lin et al., 2023). LatentDE (Tran et al., 2024) for instance combines directed evolution with a smoothed space which allows to employ gradient ascent for protein sequence design in the latent space guided by a fitness predictor.

Within the diverse array of approaches to protein fitness optimization, we propose a variational perspective that is known for its efficacy in other domains like inverse problems and image reconstruction. Specifically, we integrate a generative flow matching prior that learns the distribution of protein sequence mutations from a data set of protein sequences and combine it with a fitness predictor. The integration of this predictor helps to effectively guide a sampling process that generates high-fitness samples. Moreover, we address the problem within a compressed latent space, encoding protein sequences into a latent representation that facilitates continuous optimization techniques based on gradient information. This approach enables efficient exploration of the protein fitness landscape, leveraging the latent space to perform guided sampling directly on the encoded sequences.

## 3. Method

### 3.1. Protein Optimization

Let $x \in \mathcal{V}^d$ represent a protein sequence, with dimensionality $d$ corresponding to the number of amino acids chosen from a vocabulary $\mathcal{V}$ of 20 amino acids. Protein fitness optimization seeks to find a sequence $x$ that maximizes a specific fitness metric $y := f(x)$ with $y \in \mathbb{R}$, which quantifies desired protein functionality such as stability, activity, or expression.

Therefore, we work with paired data $\mathcal{S} = \{(x_i, y_i)\}_{i=1}^{N}$ (note that $\mathcal{S}$ is only a subset of the entire data $\mathcal{S}^*$, see Table 2) and we use the parameterized convolutional neural network (CNN) $g : \mathcal{V}^d \to \mathbb{R}$ (Kirjner et al., 2023) to infer the fitness for a given sequence. Specifically, we employ $g_\phi$ and $g_{\tilde{\phi}}$ as *predictors*, which are trained on a small subset of the data (see Section 4.1) without and with graph-based smoothing, respectively. Additionally, we use $g_\psi$ as an *in-silico oracle* for the final evaluation, trained on the entire paired data set $\mathcal{S}^*$. Note that all models share an identical architecture, differing only in their respective weights which we re-use without further training.

Building on the predictive framework for estimating protein fitness, we now turn our attention to creating new sequences that may exhibit desirable properties. Generative modeling provides a powerful approach for navigating the vast sequence space, offering a data-driven way to propose candidate proteins beyond those observed in the training set. In the following, we introduce generative models and discuss how they can be leveraged to discover novel protein sequences with optimized fitness.

### 3.2. Generative Modeling

A recent class of generative models, known as diffusion models (Ho et al., 2020; Song et al., 2021; Sohl-Dickstein et al., 2015), has shown remarkable success in generating high-quality data across various domains. These models work by progressively transforming simple noise distributions into complex data distributions through a series of iterative steps. During training, noise is systematically added to the data sample $x$ at varying levels, simulating a degradation process over time $t$. The model $\epsilon_\theta$ is then tasked with learning to reverse this process by predicting the added noise $\epsilon$ for $x_t$ at each step, effectively reconstructing the original data from the noisy observations. In detail, a model is trained to predict the added noise $\epsilon$ at each step by minimizing the objective

$$\mathcal{L}(\theta) = \mathbb{E}_{x \sim \mathrm{p}(x), t \sim \mathcal{U}_{[0,1]}, \epsilon \sim \mathcal{N}(0,I)} \left[ \|\epsilon - \epsilon_\theta(x_t, t)\|^2 \right].$$

**Flow Matching.** A more recent approach to generative modeling is given by the versatile framework of flow matching (Lipman et al., 2023; Liu et al., 2022). Rather than removing noise from data samples, flow matching aims to model the velocity of the probability flow $\Psi_t$, which governs the dynamics of how one probability distribution is transformed into another over time. By learning the velocity field $u_t$ of the probability flow, the model $v_{\theta,t}$ captures the evolution of a simple base distribution at $t = 0$ into a more

complex target distribution $p(x)$ at $t = 1$, directly modeling the flow between them. Since the velocity field $u_t$ is intractable, it was shown in (Lipman et al., 2023) that we can equivalently minimize the conditional flow matching loss:

$$\min_\theta \mathbb{E}_{t,x_1,x_0} \left[ \frac{1}{2} \| v_{\theta,t}(\Psi_t(x_0)) - (x_1 - x_0) \|^2 \right], \quad (1)$$

where $t \sim \mathcal{U}_{[0,1]}$, $x_1 \sim p(x)$ and $x_0 \sim \mathcal{N}(0, I)$, and the conditional flow is given by $\Psi_t(x_0) = (1-t)x_0 + tx_1$. Once trained, samples can be generated by numerical integration of the corresponding neural ordinary differential equation (ODE) with $t \in [0, 1]$:

$$\frac{\mathrm{d}}{\mathrm{d}t} \Psi_t(x) = v_{\theta,t}(\Psi_t(x)). \quad (2)$$

Our aim is to learn a flow-based generative model that approximates the distribution of sequence variants of a protein $p(x)$. We then seek to leverage this model in order to generate sequence variants with high protein fitness through guidance, as explained in the following.

### 3.3. Classifier guidance

Diffusion and flow-based methods can be guided by the log-likelihood gradient of an auxiliary classifier during generation, which enables conditional sampling (Dhariwal & Nichol, 2021). In detail, our goal is to sample from a conditional distribution $p(x|g_\phi(x) = y)$, where $g_\phi$ represents the predictor and $y$ denotes the desired fitness value. We therefore adopt the framework of classifier guidance which allows for a decomposition of $p(x|y) \sim p(y|x)p(x)$. Thus, we guide the sampling process by using the gradient of the predictor $g_\phi(x)$ with respect to the input sequence $x$. By introducing this gradient into the generative process, we can bias the sampling trajectory towards regions of the distribution that are more likely to yield sequences with the desired fitness value. The velocity field $v_{\theta,t}$ in the generative framework is modified to incorporate this guidance, yielding the following variational update:

$$v_{\theta,t}(x|y) = v_{\theta,t}(x) + \alpha_t \nabla_x \log p(y|x), \quad (3)$$

where $\nabla_x \log p(y|x) \sim -\frac{1}{2} \nabla_x \| g_\phi(x) - y \|^2$ represents the gradient of the log-likelihood of a sequence $x$ having desired fitness $y$ and $\alpha_t$ is a scheduler dependent constant (Zheng et al., 2023). Given the goal to maximize the fitness value of generated sequences, one could also set the gradient of the log-likelihood to $\frac{1}{2} \nabla_x \| g_\phi(x) \|^2$, or a similar suitable form. Both approaches perform very similar in practice. In this work, we chose the first version to demonstrate the possibility of steering sequences toward specific fitness values.

For the classifier guidance we use the trained CNN-based predictors $g_\phi$ and the smoothed $g_{\tilde{\phi}}$ from (Kirjner et al.,

2023). Note that for guiding the process towards the highest fitness, $y$ is simply set to 1, which represents the highest fitness in the normalized fitness spectrum.

### 3.4. Latent space representation

So far, we introduced a general framework that in theory allows for sampling from a learned distribution of protein sequences. However, these sequences represent proteins that are combinations of a discrete set of amino acids. As a result, the underlying distribution of these sequences is likely sparse and complex, making it difficult to approximate or directly sample from in its original form. To overcome this limitation, we operate in an embedded latent space, where protein sequences are encoded as continuous representations. This is achieved by a VAE framework (Kingma & Welling, 2022), which maps discrete sequences to a continuous latent space through an encoder $\mathcal{E} : \mathcal{V}^d \mapsto \mathbb{R}^l$ and reconstructs them back to the original sequence space using a decoder network $\mathcal{D} : \mathbb{R}^l \mapsto \mathbb{R}^{d \times |\mathcal{V}|}$, where $l << d$. Because of the discrete nature of amino acid tokens, the decoder produces logits, which are then mapped to tokens via an argmax operation. The training objective of the employed $\beta$-VAE (Higgins et al., 2017) is given by the weighted Evidence Lower Bound (ELBO):

$$\min_{\nu,\mu} \mathbb{E}_{z \sim q_\mu(z|x)} - \log p_\nu(x|z) + \beta \mathrm{KL}(q_\mu(z|x) \| p(z)), \quad (4)$$

where due to the discrete tokens $-\log p_\nu(x|z)$ simplifies to the cross-entropy loss in our case.

Note that while the VAE is trained with variational inference, VLGPO goes further by introducing additional generative modeling components: flow matching as described in Section 3.2 and classifier-guided sampling in Section 3.3. We show how these extensions refine the generation and help to better control it.

### 3.5. Variational Latent Generative Protein Optimization

From here on we will use the building blocks introduced earlier to describe our proposed VLGPO approach. We start by using our respective sequence data $x \sim p(x)$ to train a VAE with encoder $\mathcal{E}$ and decoder $\mathcal{D}$ that compresses the higher-dimensional discrete protein sequences into a continuous latent space representation $z \sim p(z|x)$. In order to model and sample from the learned latent space in an effective way, we train a flow matching model $v_{\theta,t}$ to learn the probability flow dynamics in the latent space. This network captures the transformation between a simple base distribution (e.g., Gaussian noise) at $t = 0$ and the complex latent distribution of protein sequences at $t = 1$. By integrating the learned flow from $t \in [0, 1]$, we can efficiently generate new latent representations $z_1$, which can subsequently be decoded back into protein sequences via the VAE decoder, resulting in $x \sim p_\nu(x|z_1) = \mathcal{D}(z_1)$, similar to (Esser et al., 2024).

**Algorithm 1** VLGPO sampling

**Require:** $K, J, y, \alpha_t$
1: Initialize $z_0 \sim \mathrm{p}_0(z) := \mathcal{N}(0, I)$
2: Set step size $\Delta t \leftarrow \frac{1}{K}$
3: **for** $k = 0$ to $K - 1$ **do**
4:     $t \leftarrow k \cdot \Delta t$
5:     $z'_t \leftarrow z_t + \Delta t\, v_{\theta,t}(z_t)$
6:     **for** $j = 0$ to $J - 1$ **do**
7:       $\hat{z}_1 \leftarrow z'_t + (1 - t - \Delta t)v_{\theta,t}(z'_t)$
8:       $z'_t \leftarrow z'_t - \frac{\alpha_t}{2}\nabla_{z'_t}\|g_\phi(\mathcal{D}(\hat{z}_1)) - y\|^2$
9:     **end for**
10:    $z_{t+\Delta t} \leftarrow z'_t$
11: **end for**
12: return $x = \mathcal{D}(z_1)$

**Sampling from the posterior.** Instead of merely sampling from the sequence distribution $\mathrm{p}(x)$, we also seek to generate high-fitness sequences. To achieve this, we condition the sampling process on a fitness score $y$, ideally set to the maximum value $y = 1$, which is estimated by a predictor $\hat{y} = g_\phi(x)$.

Although incorporating the likelihood term into the gradient, as shown in Equation (3), may appear straightforward, it is actually challenging because it becomes intractable due to the time-dependent nature of the diffusion/flow model. Moreover, explicitly including this term can push the generated samples off the current manifold related to time point $t$ (Chung et al., 2022). Inspired by (Chung et al., 2023; Ben-Hamu et al., 2024), we employ a scheme that evaluates the likelihood at $\hat{x}_1$, while the gradient of the likelihood is calculated at $x_t$, which has the effect of constraining the update to the same manifold. Therefore, at each of the $K$ sampling steps the model estimates $\hat{z}_1$ (Line 7, Algorithm 1) which is decoded to $\hat{x}_1$, the denoised version of the current decoded sample $x_t$, using the learned flow model $v_{\theta,t}$. The likelihood is then evaluated at $\hat{x}_1$ to reflect the target data distribution. However, the gradient of the likelihood, which guides the generative process, is computed with respect to $x_t$, leading to backpropagation through $v_{\theta,t}$. Furthermore, as the predictor $g_\phi$ was trained in sequence space, our likelihood function changes to

$$\nabla_x \log \mathrm{p}(y|x_1) \sim \frac{1}{2}\nabla_{z'_t}\|g_\phi(\mathcal{D}(\hat{z}_1)) - y\|^2, \quad (5)$$

where we additionally have to backpropagate through the decoder. The pseudocode of our VLGPO can be found in Algorithm 1, a detailed visualization of the sampling scheme is reported in Figure 2. The hyperparameters $J$ and $\alpha_t$ denote the number of gradient descent steps on the likelihood and the guidance strength, respectively.

## 4. Experiments

### 4.1. Data Sets

We adopt the medium and hard protein optimization benchmarks on AAV and GFP from (Kirjner et al., 2023). Given the full data set $\mathcal{S}^*$, the task difficulty is determined by *(i)* the fitness percentile range of the considered sequences ($20 - 40$ for medium, $< 30$ for hard) and *(ii)*, the required gap of mutations to reach any sequence of the 99[th] fitness percentile of $\mathcal{S}^*$ (a gap of 6 mutations for medium, 7 mutations for hard), see Table 1.

*Table 1.* Task definition.

| Task | Range % | Gap |
|---|---|---|
| Medium | 20-40 | 6 |
| Hard | < 30 | 7 |

Together, this results in four different tasks described in Table 2, each of which only sees a limited number of $N$ sequences in a limited fitness range. The idea of protein fitness optimization is to enhance these sequences to higher, previously unseen fitness values. The setting reflects realistic scenarios in terms of data set sizes. The diversity (Appendix A.2) in the training data sets of the four tasks is quite high: for GFP (medium) it is 14.5, for GFP (hard) 16.3, for AAV (medium) 15.9, and for AAV (hard) 18.4. On the other hand, the diversity within the top-performing (99[th] percentile) sequences of the full data set $\mathcal{S}^*$ is 4.73 for GFP and 5.23 for AAV.

*Table 2.* GFP and AAV data sets with number of data samples $N$, median normalized fitness scores and fitness range.

| Task | $N$ | Fitness ↑ | Fitness Range |
|---|---|---|---|
| GFP Medium | 2828 | 0.09 | $[0.01, 0.62]$ |
| GFP Hard | 2426 | 0.01 | $[0.0, 0.1]$ |
| AAV Medium | 2139 | 0.32 | $[0.29, 0.38]$ |
| AAV Hard | 3448 | 0.27 | $[0.0, 0.33]$ |

For in-silico evaluation of the generated sequences with the oracle $g_\psi$, we use the median normalized fitness, diversity and novelty following (Jain et al., 2022), see Appendix A.2. The oracle was trained on the complete DMS data with 56,086 mutants for GFP and 44,156 mutants for AAV. While diversity and novelty are reported in evaluation, no definitive higher or lower values are considered superior for a sequence. Note that $y_{\min}$ and $y_{\max}$ from the entire data set $\mathcal{S}^*$ are used for both GFP and AAV to normalize fitness scores to $[0, 1]$.

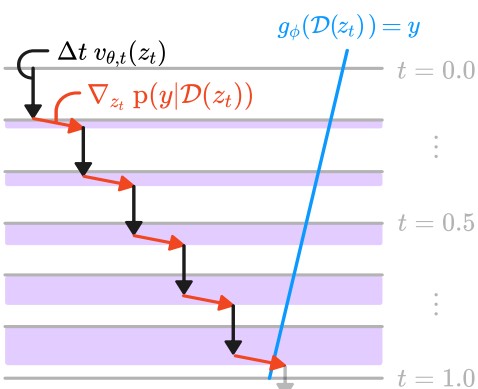 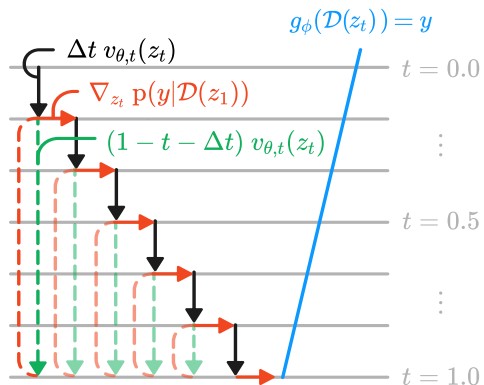

*Figure 2.* Schematic depiction of classifier guidance, with $J = 1$ and $K = 6$. Grey lines represent the latent manifolds at different time steps $t$, the blue line marks the trajectory of the maximum likelihood. Solid arrows indicate how the latent evolves over time. **Left: Naive guidance** with likelihood gradients $\nabla_{z_t}$ computed directly at $z_t$ pushes the sample off the manifold. This error accumulates, as indicated by the purple regions. **Right: Guidance with manifold constraint**, as employed in VLGPO (Algorithm 1), converges to a valid sequence with fitness $y$. Solid arrows again denote the evolution of the latent, dashed arrows indicate the flow posterior sampling scheme that ensures the latent stays on the manifold when applying the likelihood gradient.

### 4.2. Implementation Details

For *training*, we start by learning the VAE to compress the sequence token input space ($d = 28$ and $d = 237$ for AAV and GFP) to $l = 16$ and $l = 32$. A learning rate of $0.001$ with a convolutional architecture and $\beta \in \{0.01, 0.001\}$ for AAV and GFP is used for training the encoder $\mathcal{E}$ and decoder $\mathcal{D}$ in Equation (4), also see Appendix B.2. To learn the prior distribution of embedded latent sequences $z = \mathcal{E}(x)$ using flow matching, the 1D CNN commonly used for denoising diffusion probabilistic models (DDPMs)[2] is employed. A learning rate of $5e\text{-}5$ and a batch size of $1024$ were used to train $v_{\theta,t}$ for 1000 epochs. All VAE and flow matching models are always trained under the limited-data setting as listed in Table 2.

At *inference time*, we follow the procedure outlined in Algorithm 1 to generate samples. The predictors $g_\phi$ and $g_{\tilde{\phi}}$ (for details, see Appendix A.1) are re-used without any further training (Kirjner et al., 2023). We start from $z_0 \sim \mathcal{N}(0, I)$ and use $K = 32$ ODE steps to integrate the learned flow until we obtain $z_1$. To optimize sequence fitness, the condition $y = 1$ is selected for all samples. The parameters $\alpha_t$ and $J$ are determined via a hyperparameter search, as shown in Figure 3 and discussed in detail in Appendix B.4. Ultimately, they modulate the trade-off between increasing fitness and maintaining diversity. After generating 512 samples $z_1$ to encourage sampling from the entire learned distribution, they are decoded using $x = \mathcal{D}(z_1)$. Potential duplicates are then filtered out, and the top-$k$ ($k = 128$) samples, ranked by the predictor ($g_\phi$ or $g_{\tilde{\phi}}$, respectively), are selected. Note that also the predictors $g_\phi$ and $g_{\tilde{\phi}}$ for

each setting, used for classifier guidance and for ranking the samples, are trained only on the data sets listed in Table 2.

For *evaluation* of the generated samples, we use the oracle $g_\psi$ as in-silico fitness estimate, see Appendix A.1. The oracle is directly sourced from (Kirjner et al., 2023) and is the only model that was trained using the entire data $\mathcal{S}^*$. For sampling following Algorithm 1, we use $g_\phi$ and compare VLGPO to GWG (which also uses the same trained predictor), and the identical smoothed predictor $g_{\tilde{\phi}}$ in VLGPO to GGS (Kirjner et al., 2023). We benchmark against the respective baselines, namely GFlowNets (GFN-AL) (Jain et al., 2022), model-based adaptive sampling (CbAS) (Brookes et al., 2019), greedy search (AdaLead) (Sinai et al., 2020), Bayesian optimization (BO-qei) (Wilson et al., 2017), conservative model-based optimization (CoMS) (Trabucco et al., 2021) and proximal exploration (PEX) (Ren et al., 2022). Moreover, we investigate the performance of the recently introduced gradient-guided walk-jump sampling algorithm (gg-dWJS), which extends the walk-jump sampling framework originally developed for antibody sequence design (Ikram et al., 2024; Frey et al., 2023). Because the available source code was not directly compatible with the data sets in Table 2, we adapted our existing model architectures for implementation. We then performed a grid search over sampling parameters, followed by top-$k$ sampling for each task, to ensure a fair comparison.

### 4.3. Results

We compare VLGPO as illustrated in Algorithm 1 for the four tasks in Table 2 by averaging over five seeds and computing the median normalized fitness, diversity and novelty (see Appendix A.2). The results reported in Table 3 and Ta-

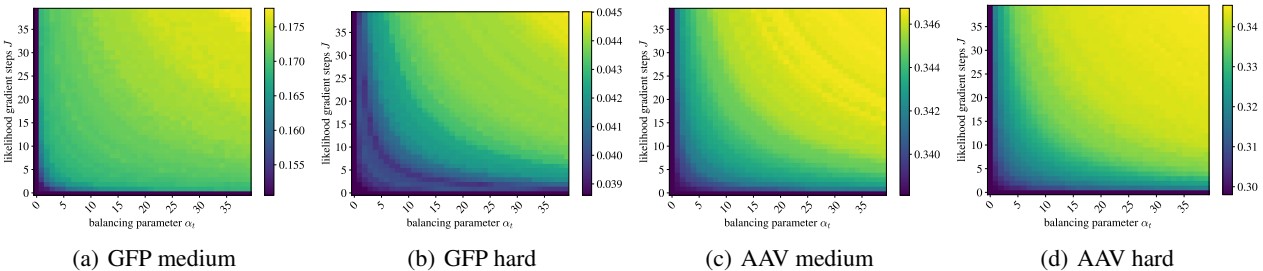

(a) GFP medium        (b) GFP hard        (c) AAV medium        (d) AAV hard

*Figure 3.* Grid search for median fitness depending on sampling parameters $\alpha_t$ and $J$ for the different tasks using the predictor $g_\phi$. In general, higher values of $\alpha_t$ and $J$, corresponding to strong classifier guidance, yield higher predicted fitness values.

ble 4 for GFP and AAV respectively, demonstrate the fitness improvement of VLGPO with both predictors ($g_\phi$ and $g_{\tilde{\phi}}$) over all other benchmarked methods. In particular, VLGPO shows clear fitness improvement over GWG (which uses the same predictor $g_\phi$) and GGS (which uses the same predictor $g_{\tilde{\phi}}$). Moreover, the performance difference between our method and GWG (case of using non-smoothed predictor $g_\phi$) further highlights the robustness of our approach in limited data regimes and supports the advantage of the guided flow matching prior in the latent space.

When it comes to diversity and novelty, there is no clear definition of what is better – higher or lower scores (Kirjner et al., 2023). In general, we observe that reported values of VLGPO are in line with competing methods. Moreover, the variations between different methods in terms of diversity and novelty are significantly larger in GFP than in AAV. That discrepancy may arise due to GFP sequences being longer, thus representing a sparser and higher-dimensional search space that makes the protein harder to optimize. Interestingly, the sequences in the 99[th] fitness percentile, which are on the top end of high-fitness sequences, show a diversity of 4.73 for GFP and 5.23 for AAV, which are approximately comparable to our results in Table 3 and Table 4, except for AAV hard. In practice, the smoothed predictor $g_{\tilde{\phi}}$ reduces the diversity within the generated sequences for both GGS and VLGPO. Additionally, more samples tend to collapse to the same decoded sequence, likely due to the smoother, less distinct gradients.

While gg-dWJS is conceptually similar to VLGPO, its performance on both GFP and AAV (hard) does not fully align with our findings. We hypothesize two main reasons for this discrepancy. First, the GFP data set (Table 2) is relatively small and limited to mutations of a single protein, posing a challenge for methods that rely on one-hot encoding. Although gg-dWJS performs well on AAV, the one-hot-encoded space becomes extremely sparse for the longer GFP sequences ($d = 237$), causing many generated samples to collapse onto identical sequences. Second, the discriminator model acting as the guidance for gg-dWJS

(analogous to our predictor $g_\phi$) is trained in the noisy one-hot-encoded space of sequences with a single noise level, which limits its predictive capabilities.

### 4.4. Fitness Extrapolation

Table 2 illustrates the limited fitness range of the sequences in each task, whereas protein fitness optimization aims to sample sequences with higher fitness values. Because the available data set is small, extrapolating to these high-fitness sequences during sampling becomes especially challenging. The following experiment examines how much the oracle-evaluated fitness $y_{gt}$ (provided by $g_\psi$) deviates from the target fitness $y$, which is used as an input to the sampling in Algorithm 1. Thereby, we compare two approaches: our variational method VLGPO and a directly learned posterior $p(x|y)$ via fitness-conditioned flow matching $v_{\theta,t}(z_t, y)$. The learned posterior does not need the predictor $g_\phi$ for classifier guidance, as it learns the conditional distribution in the latent space end-to-end. For a fair comparison, the raw output without top-$k$ sampling is applied. The results are depicted in Figure 4 for GFP and AAV hard.

Due to the sparse data availability and the limited fitness range, the evaluated fitness $y_{gt}$ cannot be expected to precisely follow the required fitness $y$. Nevertheless, the results in Figure 4 clearly demonstrate the advantage of classifier guidance in VLGPO, whereas the direct posterior fails to effectively exploit the given fitness condition. This gap is most evident in the GFP hard task, where the training data's fitness range is only in $[0.0, 0.1]$, making it difficult for the learned posterior to extrapolate to higher fitness values. In contrast, VLGPO leverages the additional gradient information from the predictor $g_\phi$, thereby overcoming these limitations in higher-fitness regions. This highlights the advantages of using a separate classifier for guidance in domains where classifier-free guidance struggles to produce high-fitness sequences.

*Table 3.* GFP optimization results. Best score for fitness in **bold**, second-best underlined, and the results for our method (VLGPO) are highlighted in grey.

| Method | Medium difficulty | | | Hard difficulty | | |
|---|---|---|---|---|---|---|
| | Fitness ↑ | Diversity | Novelty | Fitness ↑ | Diversity | Novelty |
| GFN-AL | $0.09 \pm 0.1$ | $25.1 \pm 0.5$ | $213 \pm 2.2$ | $0.1 \pm 0.2$ | $23.6 \pm 1.0$ | $214 \pm 4.2$ |
| CbAS | $0.14 \pm 0.0$ | $9.7 \pm 1.1$ | $7.2 \pm 0.4$ | $0.18 \pm 0.0$ | $9.6 \pm 1.3$ | $7.8 \pm 0.4$ |
| AdaLead | $0.56 \pm 0.0$ | $3.5 \pm 0.1$ | $2.0 \pm 0.0$ | $0.18 \pm 0.0$ | $5.6 \pm 0.5$ | $2.8 \pm 0.4$ |
| BOqei | $0.20 \pm 0.0$ | $19.3 \pm 0.0$ | $0.0 \pm 0.0$ | $0.0 \pm 0.5$ | $94.6 \pm 71$ | $54.1 \pm 81$ |
| CoMS | $0.00 \pm 0.1$ | $133 \pm 25$ | $192 \pm 12$ | $0.0 \pm 0.1$ | $144 \pm 7.5$ | $201 \pm 3.0$ |
| PEX | $0.47 \pm 0.0$ | $3.0 \pm 0.0$ | $1.4 \pm 0.2$ | $0.0 \pm 0.0$ | $3.0 \pm 0.0$ | $1.3 \pm 0.3$ |
| gg-dWJS | $0.55 \pm 0.1$ | $52.3 \pm 3.4$ | $16.3 \pm 5.7$ | $0.61 \pm 0.1$ | $68.0 \pm 5.6$ | $44.8 \pm 47$ |
| GWG | $0.10 \pm 0.0$ | $33.0 \pm 0.8$ | $12.8 \pm 0.4$ | $0.0 \pm 0.0$ | $4.2 \pm 7.0$ | $7.6 \pm 1.1$ |
| VLGPO, predictor $g_\phi$ | $\mathbf{0.87 \pm 0.0}$ | $4.31 \pm 0.1$ | $6.0 \pm 0.0$ | $\underline{0.75 \pm 0.0}$ | $3.1 \pm 0.2$ | $6.0 \pm 0.0$ |
| GGS | $0.76 \pm 0.0$ | $3.7 \pm 0.2$ | $5.0 \pm 0.0$ | $0.74 \pm 0.0$ | $3.6 \pm 0.1$ | $8.0 \pm 0.0$ |
| VLGPO, smoothed $g_{\tilde{\phi}}$ | $\underline{0.84 \pm 0.0}$ | $2.06 \pm 0.1$ | $5.0 \pm 0.0$ | $\mathbf{0.78 \pm 0.0}$ | $2.5 \pm 0.2$ | $6.0 \pm 0.0$ |

*Table 4.* AAV optimization results. Best score for fitness in **bold**, second-best underlined, and the results for our method (VLGPO) are highlighted in grey.

| Method | Medium difficulty | | | Hard difficulty | | |
|---|---|---|---|---|---|---|
| | Fitness ↑ | Diversity | Novelty | Fitness ↑ | Diversity | Novelty |
| GFN-AL | $0.20 \pm 0.1$ | $9.60 \pm 1.2$ | $19.4 \pm 1.1$ | $0.10 \pm 0.1$ | $11.6 \pm 1.4$ | $19.6 \pm 1.1$ |
| CbAS | $0.43 \pm 0.0$ | $12.7 \pm 0.7$ | $7.2 \pm 0.4$ | $0.36 \pm 0.0$ | $14.4 \pm 0.7$ | $8.6 \pm 0.5$ |
| AdaLead | $0.46 \pm 0.0$ | $8.50 \pm 0.8$ | $2.8 \pm 0.4$ | $0.40 \pm 0.0$ | $8.53 \pm 0.1$ | $3.4 \pm 0.5$ |
| BOqei | $0.38 \pm 0.0$ | $15.22 \pm 0.8$ | $0.0 \pm 0.0$ | $0.32 \pm 0.0$ | $17.9 \pm 0.3$ | $0.0 \pm 0.0$ |
| CoMS | $0.37 \pm 0.1$ | $10.1 \pm 5.9$ | $8.2 \pm 3.5$ | $0.26 \pm 0.0$ | $10.7 \pm 3.5$ | $10.0 \pm 2.8$ |
| PEX | $0.40 \pm 0.0$ | $2.80 \pm 0.0$ | $1.4 \pm 0.2$ | $0.30 \pm 0.0$ | $2.8 \pm 0.0$ | $1.3 \pm 0.3$ |
| gg-dWJS | $0.48 \pm 0.0$ | $9.48 \pm 0.3$ | $4.2 \pm 0.4$ | $0.33 \pm 0.0$ | $14.3 \pm 0.7$ | $5.3 \pm 0.4$ |
| GWG | $0.43 \pm 0.1$ | $6.60 \pm 6.3$ | $7.7 \pm 0.8$ | $0.33 \pm 0.0$ | $12.0 \pm 0.4$ | $12.2 \pm 0.4$ |
| VLGPO, predictor $g_\phi$ | $\mathbf{0.58 \pm 0.0}$ | $5.58 \pm 0.2$ | $5.0 \pm 0.0$ | $0.51 \pm 0.0$ | $8.44 \pm 0.2$ | $7.8 \pm 0.4$ |
| GGS | $0.51 \pm 0.0$ | $4.0 \pm 0.2$ | $5.4 \pm 0.5$ | $\underline{0.60 \pm 0.0}$ | $4.5 \pm 0.5$ | $7.0 \pm 0.0$ |
| VLGPO, smoothed $g_{\tilde{\phi}}$ | $\underline{0.53 \pm 0.0}$ | $4.96 \pm 0.2$ | $5.0 \pm 0.0$ | $\mathbf{0.61 \pm 0.0}$ | $4.29 \pm 0.1$ | $6.2 \pm 0.4$ |

## 4.5. Ablation Studies

We conduct an ablation study on the influence of manifold constrained gradients in sampling (Line 7, Algorithm 1). The results in Table 5 and Table 6 demonstrate the improvement gained by estimating $\hat{x}_1$ to compute the gradient of the likelihood term.

Further, the performance of the directly learned posterior $p(x|y)$ was investigated. While this approach still performs well for the medium tasks, it shows a larger performance drop on GFP (hard), indicating that it cannot match the explicit likelihood guidance provided by the fitness predictor in our variational method VLGPO.

*Table 5.* Influence of manifold constrained gradient in sampling for both predictors $g_\phi$ and smoothed $g_{\tilde{\phi}}$ and directly learned posterior for GFP medium and hard tasks.

| Method | Medium Fitness ↑ | Hard Fitness ↑ |
|---|---|---|
| VLGPO, predictor $g_\phi$ | $0.87 \pm 0.0$ | $0.75 \pm 0.0$ |
| w/o manifold constraint | $0.81 \pm 0.0$ | $0.73 \pm 0.0$ |
| learned posterior | $0.83 \pm 0.1$ | $0.44 \pm 0.1$ |
| VLGPO, smoothed $g_{\tilde{\phi}}$ | $0.84 \pm 0.0$ | $0.78 \pm 0.0$ |
| w/o manifold constraint | $0.84 \pm 0.0$ | $0.67 \pm 0.1$ |

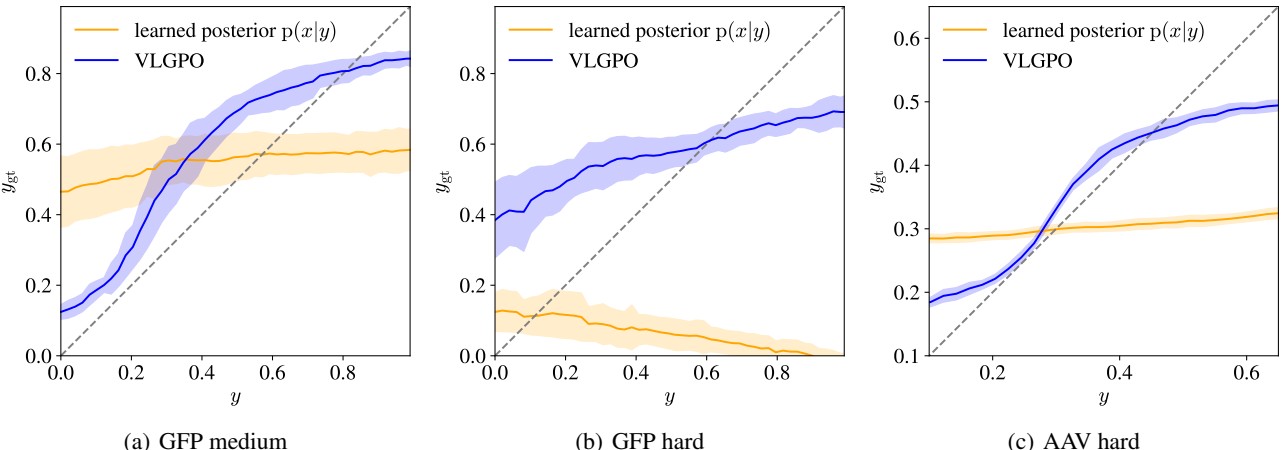

(a) GFP medium        (b) GFP hard        (c) AAV hard

*Figure 4.* Comparing evaluated fitness $y_{gt}$ from the oracle $g_\psi$ with required fitness $y$ using the directly learned posterior model (in the same latent space) and our variational approach VLGPO.

*Table 6.* Influence of manifold constrained gradient in sampling for both predictors $g_\phi$ and smoothed $g_{\tilde{\phi}}$ and directly learned posterior for AAV medium and hard tasks.

| Method | Medium Fitness ↑ | Hard Fitness ↑ |
|---|---|---|
| VLGPO, predictor $g_\phi$ | 0.58 ± 0.0 | 0.51 ± 0.0 |
| w/o manifold constraint | 0.55 ± 0.0 | 0.47 ± 0.0 |
| learned posterior | 0.55 ± 0.0 | 0.44 ± 0.0 |
| VLGPO, smoothed $g_{\tilde{\phi}}$ | 0.53 ± 0.0 | 0.61 ± 0.0 |
| w/o manifold constraint | 0.52 ± 0.0 | 0.58 ± 0.0 |

## 5. Discussion

We present VLGPO, a variational approach for protein fitness optimization that enables posterior sampling of high-fitness sequences. Our method operates in a learned smoothed latent space and learns a generative flow matching prior that imposes a natural gradient regularization, removing the need for extra smoothing. Additionally, it incorporates a likelihood term through manifold-constrained gradients that helps guiding the sampling process towards high-fitness regions. VLGPO achieves clear fitness improvements, with respect to the fitness of training sequences, but especially when compared to all baseline and competing methods.

The variational framework offers a versatile approach, as its modular design allows replacing any of its components as needed, such as the prior model, the predictor, or the architectures. Future work could explore using embeddings from pretrained protein language models (pLMs) instead of the VAE, since such embeddings provide a more expressive latent representation. However, this would require finetuning the decoder to ensure faithful sequence reconstruction, which can be prone to overfitting given the limited size of employed data sets.

A limiting factor of our method lies in its hyperparameter tuning requirements. We observe that hyperparameter selection becomes more critical for challenging tasks such as GFP (hard), while it remains stable and robust for other tasks. Additionally, the restriction of the benchmark to only AAV and GFP is a limitation that should be addressed in future work. Conceptually, VLGPO as well as competing approaches can be extended to other proteins from FLIP (Dallago et al., 2021) or ProteinGym (Notin et al., 2023).

Another important point is our reliance on in-silico evaluation, where we follow (Kirjner et al., 2023) in using a trained oracle as the ground truth. This oracle was trained on a significantly larger data set than the tasks presented in Table 2 and can therefore be expected to serve as a good estimator. Nevertheless, actual experimental validation could provide valuable insights into the applicability of VLGPO. A complementary direction may be to design more idealised, synthetic in-silico benchmarks that are nevertheless good proxies for protein design (Stanton et al., 2024). Moreover, additional metrics such as folding confidence or structural stability could be added to obtain a more complete perspective beyond the fitness score (Johnson et al., 2023).

## Acknowledgements

This work was supported by the University Research Priority Program (URPP) Human Reproduction Reloaded of the University of Zurich.

## Impact Statement

This paper presents a computational method for protein fitness optimization. There are many potential societal consequences of our work, none of which we feel must be specifically highlighted here.

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

## A. Additional Methods

### A.1. Fitness predictor and oracle

The fitness predictors $g_\phi$ and $g_{\tilde{\phi}}$, as well as the oracle $g_\psi$ are directly re-used from (Kirjner et al., 2023) without any modifications. The authors use the identical 1D CNN architecture with 157k learnable parameters for all networks, which maps a one-hot encoded protein sequence to a scalar value representing the predicted fitness. Our method VL-GPO allows to replace all of these components by re-trained models or alternative architectures, although it is known that simple CNNs can be competitive in such data-scarce settings (Dallago et al., 2021).

Since (Kirjner et al., 2023) does not explicitly report the final performance of the in-silico oracle, we compute the Mean Squared Error (MSE) on a subset of 512 randomly selected samples from the ground truth data set $\mathcal{S}^*$ to estimate its reliability. The oracle's predictions closely follow the target fitness values, resulting in MSE values of 0.012240 for GFP and 0.002758 for AAV.

### A.2. Metrics

For the evaluation of the generated sequences, we use median fitness, diversity and novelty as described in (Kirjner et al., 2023; Jain et al., 2022). The sequences are assessed by the oracle $g_\psi$ that was trained on the full data set $\mathcal{S}^*$ with minimum and maximum fitness values $y_{\min}$ and $y_{\max}$. The median normalized fitness of sampled sequences $x \in \mathcal{X}$ is computed using

$$\mathrm{median}\Big(\Big\{\frac{g_\psi(x) - y_{\min}}{y_{\max} - y_{\min}} \ : \ x \in \mathcal{X}\Big\}\Big).$$

Diversity is defined as the median similarity within the set of generated sequences by

$$\mathrm{median}\Big(\big\{\mathrm{dist}(x, x') : x, x' \in \mathcal{X}, \ x \neq x'\big\}\Big),$$

using the Levenshtein distance as we are evaluating discrete sequences. Finally, novelty considers the minimum distance of the generated sequences $x \in \mathcal{X}$ to any of the sequences in the respective data set $\mathcal{S}$ that the generative flow matching prior was trained on (see Table 2). Therefore, it is computed using

$$\mathrm{median}\Big(\big\{\min_{\substack{\hat{x} \in \mathcal{S} \\ \hat{x} \neq x}}\{\mathrm{dist}(x, \hat{x}) \ : \ x \in \mathcal{X}\}\big\}\Big).$$

## B. Additional Results

### B.1. Fitness Extrapolation

The experiments on fitness extrapolation of generated sequences shown in Section 4.4 are complemented by the

additional task of AAV (medium) in Figure 5. It supports the finding that the classifier guidance in combination with the generative prior in our variational approach VLGPO yields sequences whose predicted fitness $y_{\mathrm{gt}}$ follow the conditioned fitness $y$ more closely than the directly learned posterior model.

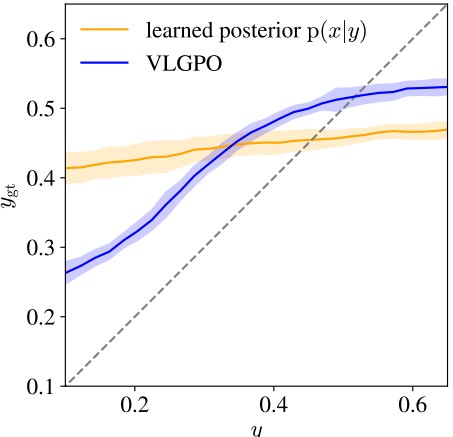

*Figure 5.* AAV medium

### B.2. Variational Autoencoder

The VAE embeds sequences into a latent space that yields continuous gradients for sampling in Algorithm 1. Empirically, we choose $l = 16$ and $l = 32$ for AAV and GFP, since further compression reduces the decoder $\mathcal{D}$'s reconstruction accuracy and harms sequence generation. Moreover, as the medium and hard tasks are defined by a minimum number of mutations from the 99th fitness percentile of $\mathcal{S}^*$, a sufficiently large latent dimensionality is required to retain all relevant information.

We train a VAE for each task to achieve at least $80\%$ reconstruction accuracy on a validation subset, balancing the latent prior with $\beta$. Table 7 shows these results. The lower accuracy in AAV compared to GFP is due to larger effects of mutations on shorter proteins (sequence length $d$). Since the data sets (see Table 2) are very small, the Kullback-Leibler (KL) divergence in Equation (4) is crucial to ensure a Gaussian latent distribution. We can also sample from the VAE and evaluate predicted fitness via $g_\psi$ (Table 7), although median normalized fitness is lower than in Table 2. This is expected, given the sparse latent space and the "hole" problem of VAEs (Rezende & Viola, 2018).

### B.3. Unconditional Sampling

In Algorithm 1 one can recover the unconditional sampling case by setting $\alpha_t = 0$ and $J = 0$, thereby sampling exclusively from the learned generative flow matching prior. As a result, the classifier guidance in the likelihood term does not

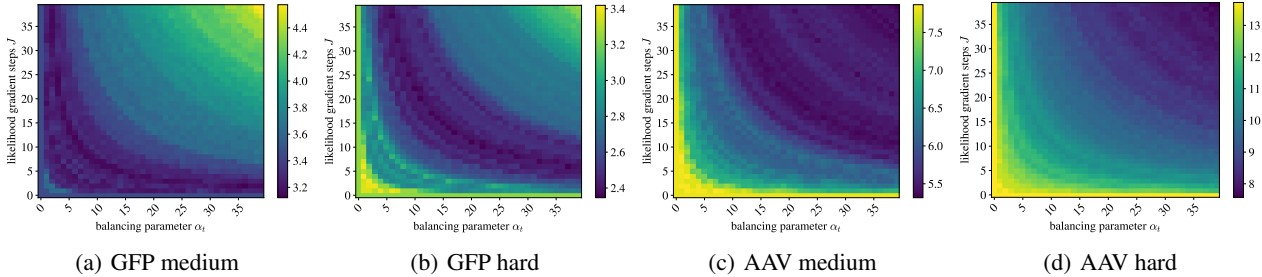

|     |     |     |     |
| :-: | :-: | :-: | :-: |
| (a) GFP medium | (b) GFP hard | (c) AAV medium | (d) AAV hard |

*Figure 6.* Grid search for diversity depending on sampling parameters $\alpha_t$ and $J$ for the different tasks.

*Table 7.* VAE reconstruction and sampling results.

| Task | Reconstruction Accuracy ↑ | Fitness ↑ |
| :--- | :---: | :---: |
| GFP Medium | 97.0% | -0.08 |
| GFP Hard | 96.8% | -0.21 |
| AAV Medium | 80.4% | 0.28 |
| AAV Hard | 87.8% | 0.23 |

impact the sampling procedure. The median fitness values of the generated samples, evaluated as usual by the oracle $g_\psi$, are shown in Table 8. Note that, in the absence of the predictor $g_\phi$ during VLGPO sampling, the flow matching model effectively serves only as a prior without any fitness conditioning, so no post-processing with top-$k$ sampling is applied for the sake of solely analyzing the prior model.

As shown in Table 8, the median fitness is lower than in Table 3 and Table 4, both of which use VLGPO with classifier guidance $g_\phi$. However, it exceeds the median fitness of the data sets $\mathcal{S}$ from Table 2, likely due to limited training data and the lack of fitness information of the flow matching prior. Moreover, the model may exhibit a mode-seeking tendency, concentrating on modes that are easier to model. While the averaged novelty aligns with expectations, diversity is much higher in the unconditional scenario, since no classifier guidance steers samples towards higher-fitness modes.

*Table 8.* Unconditional optimization results ($\alpha_t = 0$, $J = 0$).

| Task | Fitness ↑ | Diversity | Novelty |
| :--- | :---: | :---: | :---: |
| GFP Medium | $0.22 \pm 0.1$ | $18.9 \pm 2.5$ | $6.2 \pm 0.4$ |
| GFP Hard | $0.42 \pm 0.1$ | $14.2 \pm 1.1$ | $7.0 \pm 0.0$ |
| AAV Medium | $0.38 \pm 0.0$ | $11.8 \pm 0.1$ | $6.0 \pm 0.0$ |
| AAV Hard | $0.28 \pm 0.0$ | $15.6 \pm 0.2$ | $8.0 \pm 0.0$ |

**B.4. Sampling Parameters**

The determination of the hyperparameters $\alpha_t$ and $J$ in VL-GPO sampling is performed through a grid search across dif-

ferent tasks. We keep $\alpha_t$ constant for all steps $t$. In addition to Figure 3, the resulting heatmaps for these two parameters on diversity are shown in Figure 6. Overall, combining the findings on predicted fitness and diversity it seems to be a heuristic tradeoff to balance these metrics, unlike for GFP (medium), which allows to choose both parameters high. Likewise, for AAV (hard) and (medium) there appears to be a broad range from which suitable parameter choices for $\alpha_t$ and $J$ can be made. Only GFP (hard) displays substantially compromised diversity very quickly (note the limited numerical range displayed in the colorbar), hence we choose significantly lower values $\alpha_t = 0.02$ and $J = 5$. For the other tasks, we use the hyperparameter settings obtained from the grid search experiments, i.e., $\alpha_t \in \{0.97, 1.2, 0.56\}$ and $J \in \{39, 19, 37\}$ for AAV (medium), AAV (hard) and GFP (medium). Nevertheless, we do not observe major differences if these hyperparameters are adjusted slightly.

Finally, the influence of the number of ODE steps, which directly corresponds to the sampling steps $K$, is examined. This is shown in Figure 7, with the resulting fitness on the left and diversity on the right. The fitness initially exhibits an overshooting behavior, accompanied by a decrease in diversity, but this effect stabilizes after approximately 10 sampling steps. Beyond this point, both metrics remain relatively constant and stable with respect to the number of sampling steps $K$, which is generally the desired behavior.

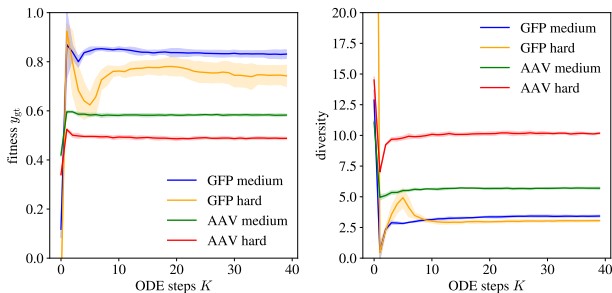

*Figure 7.* Median fitness (left) and diversity (right) for all four tasks depending on employed ODE steps $K$ in sampling.

