# OpenReview forum: "A Variational Perspective on Generative Protein Fitness Optimization"
_ICML.cc/2025/Conference — ICML 2025 poster_

### Official Review · Reviewer_Lapo · 2025-02-21

**Overall Recommendation:** 2

**Summary:**

The paper focus on the problem of protein fitness optimization to find new variants with enhanced fitness, facing challenges such as a vast search space and discrete protein sequences. The introduced Variational Latent Generative Protein Optimization (VLGPO) is a variational framework that enables posterior sampling of protein sequences based on desired fitness levels by combining a learned prior and fitness predictor. They first train a VAE on the sequence space to compress sequence into continuous latent representation and then train a flow matching model to learn a generative prior for the latent space. The fitness predictor is used for classifier-guidance of flow matching generation. VLGPO demonstrates strong performance on AAV and GFP benchmarks in medium and high difficulty tasks with limited data, achieving clear fitness improvements over baselines.  The method has limitations in hyperparameter tuning, especially for challenging tasks like GFP (hard), and relies on in-silico evaluation with a trained oracle as ground truth, where experimental validation could provide more insights into its applicability.

**Claims And Evidence:**

All contribution points are supported in the paper, except some concerns regarding evaluation. See below.

**Essential References Not Discussed:**

[1] Notin, Pascal, et al. "Proteingym: Large-scale benchmarks for protein fitness prediction and design." Advances in Neural Information Processing Systems 36 (2023): 64331 - 64379.

[2] Ouyang-Zhang, Jeffrey, et al. "Predicting a protein's stability under a million mutations." Advances in Neural Information Processing Systems 36 (2023): 76229 - 76247.

**Experimental Designs Or Analyses:**

I have serious doubts about the soundness of the experimental design and the validity of the results presented.
1. All evaluations are conducted on relatively small benchmarks, containing fewer than 3,500 mutants. This narrow scope casts significant doubt on the reliability of the evaluation. Given that it only represents a small fraction of the vast possible mutation landscape, the generalizability of the findings is severely compromised. Moreover, the predictors and oracles, trained and evaluated on these limited datasets, are also suspect. In contrast, many recent datasets, such as those in [1] and [2], have been released with 10 to 100 times more mutants, highlighting the inadequacy of the current benchmark size.
2. The dataset is split according to fitness values, with lower-value data used for training the fitness predictor and higher-value data as the generation objective. However, in real-world protein engineering, the typical approach is to design single-mutants first, followed by double-mutants and then triple-mutants. Therefore, a more appropriate split should be based on mutation depth. Additionally, accurately modeling the effects of multiple mutations remains a challenging problem, which poses a significant challenge to the accurate training of oracles and predictors.
3. There is a lack of clarity regarding the dataset used to train the flow matching and VAE models. It is not specified whether they are trained on $S^*$ or if the training data contains all mutants.
4. The achievement of high fitness values comes at a substantial cost of sacrificing diversity and novelty. This trade-off is highly undesirable.
5. How the authors perform the grid search for hyperparameters such as $\alpha_t$ and $J$ is unclear. It appears that the search is based on the final reported metrics. This raises concerns about the practical applicability of the method in real-world scenarios, as the hyperparameter tuning strategy may not be robust enough for different contexts.
6. The authors should incorporate more realistic settings and additional baselines, similar to those presented in [1] and [2].

[1] Notin, Pascal, et al. "Proteingym: Large-scale benchmarks for protein fitness prediction and design." Advances in Neural Information Processing Systems 36 (2023): 64331 - 64379.

[2] Ouyang-Zhang, Jeffrey, et al. "Predicting a protein's stability under a million mutations." Advances in Neural Information Processing Systems 36 (2023): 76229 - 76247.

**Methods And Evaluation Criteria:**

The proposed method suffers from a lack of clarity and fails to demonstrate sufficient novelty.
1. The predictors $g_\phi$, $g_\tilde{\phi}$, and the oracle $g_{\psi}$ are sourced directly from previous works. However, the paper offers no insights into their design process, the training process on the benchmark, or their accuracy levels. This absence of information leaves readers uncertain about the reliability of these predictors.
2. When training a flow matching model, the parameterization of the neural network and the choice of the time step sampling distribution significantly impact the final performance. These factors should be carefully considered as hyperparameters to be tuned. Unfortunately, the paper neglects to address this, leaving a gap in understanding how the model was optimized for optimal performance.
3. Most of the models used in the evaluation are based on CNNs. Nowadays, transformer-based models, such as ESM, have shown superiority over CNNs. This raises doubts about the reliability of the evaluation. Using outdated models may lead to an inaccurate assessment of the proposed method's performance compared to the latest advancements in the field.
4. Regarding classifier guidance, it remains unclear why the objective is to optimize towards a specific fitness value rather than maximizing the fitness value. In real-world scenarios, it is entirely possible to achieve fitness values higher than the highest ones present in the dataset. 5. The idea of applying flow matching on the latent space and guiding generation with a classifier has been proposed many times in different domains. This work seems to directly apply this method on the protein sequence design task.

**Other Comments Or Suggestions:**

See above.

**Other Strengths And Weaknesses:**

See above.

**Questions For Authors:**

See above.

**Relation To Broader Scientific Literature:**

See the summary, method and experiment parts.

**Theoretical Claims:**

N/A

---

> ### Author Rebuttal · Authors · 2025-03-27
>
> We thank the reviewer for providing feedback on our manuscript and appreciate the comments. We will also discuss related work that you mentioned.
>
> - **The predictors and the oracle are sourced directly from previous works.**
>
> Indeed, we take this from recent work from ICLR’24 [1]. As mentioned to reviewer TQyg, we will further elaborate on this.
>
> - **Parameterization of the NN and time step sampling choices for flow matching.**
>
> Our experiments have shown to be very robust against the choice of time steps used for inference, as we use a flow-based approach that is much less sensitive to hyperparameter settings than standard time-discrete diffusion models. The choice of the architecture at hand is rather generic and used as such for many other related tasks.
>
> The time steps are sampled uniformly between 0 and 1 in training, which is the standard for flow-based approaches, we saw no need to change it.
>
> - **Nowadays, transformer-based models, such as ESM, have shown superiority over CNNs.**
>
> Transformer-based models are standard in protein design, we show that guided flow matching performs well even with simpler architectures (although our network does include attention layers). More expressive models might yield further improvements. Note, previous work [1,2] has discussed that CNN-based models are competitive in limited-data settings.
>
> As noted in Sec. 5, ESM embeddings are highly expressive, and we plan to explore their potential in future work.
>
> - **Regarding classifier guidance, it remains unclear why the objective is to optimize towards a specific fitness value rather than maximizing the fitness value.**
>
> Maximizing the fitness value is our ultimate goal; however, we intended to demonstrate the expressiveness of our approach by steering sequences toward specific fitness values. In practice, we compared both formulations which performed similarly. Given that our variational framework allows for fitness maximization in the likelihood term, we will explicitly incorporate this into Eq (3). If of interest, we can provide the results in the supplementary.
>
> - **The idea of applying flow matching on the latent space and guiding generation with a classifier has been proposed many times in different domains.**
>
> We agree that classifier guidance is widely used in generative AI. However, it cannot be trivially applied to discrete sequences, and its application in latent spaces for protein fitness optimization has not been explored before, to the best of our knowledge.
>
> - **All evaluations are conducted on relatively small benchmarks which casts significant doubt on the reliability of the evaluation.**
>
> We respectfully disagree with this concern. Our evaluation protocol and benchmarks align directly with established standards from previously published work [1]. Such dataset sizes reflect realistic scenarios. Moreover, the in-silico oracle was trained on the complete DMS datasets with 56,086 mutants for GFP and 44,156 for AAV.
>
> - **The dataset is split according to fitness values... a more appropriate split should be based on mutation depth**.
>
> Both tasks *medium* and *hard* indeed are based on fitness values as well as mutation gaps. This is discussed in [1] (independent original work) and in our manuscript in Sec. 4.1 and in Tab. 6.
>
> - **There is a lack of clarity regarding the dataset used to train the flow matching and VAE models.**
>
> For each of the four tasks, the flow matching and VAE models are trained exclusively on their respective limited datasets to emulate a realistic scenario. We will explicitly clarify this in the revised manuscript.
>
> - **The achievement of high fitness values comes at a substantial cost of sacrificing diversity and novelty.**
>
> We agree that there is an inherent trade-off between achieving high fitness values and maintaining diversity, which is also reflected in our approach. Given that our method achieves a substantial improvement in fitness compared to alternative approaches, we believe the benefit outweighs the marginal drop in diversity compared to competing methods.
>
> - **How the authors perform the grid search for hyperparameters is unclear.**
>
> The plots shown in the manuscript are generated by the oracle, which are an ablation and not the source of our hyperparameters, similar to [1]. We agree, this was not explained well enough. For our parameter estimation we combine grid searches over the diversity as well as the fitness given only by the predictor. We will add the diversity and predictor fitness plots (which resemble Fig. 5) in the appendix. Ultimately it is a heuristic tradeoff between fitness and diversity, which we will further elaborate upon.
>
> - **The authors should incorporate more realistic settings and additional baselines.**
>
> See response to reviewer jm6F, last point.
>
> [1] Kirjner, Andrew, et al. "Improving protein optimization with smoothed fitness landscapes."
>
> [2] Dallago, Christian, et al. "FLIP: Benchmark tasks in fitness landscape inference for proteins."

---

> > ### Comment · Reviewer_Lapo · 2025-04-03
> >
> > Thanks for the author's detailed response. My major concern is around the evaluation settings. I totally understand the reason the authors choose a well-established benchmark for the evaluation (AAV and GFP). This is mostly because the two proteins are well studied, have abundant data and people have already developed very accurate function predictors as the authors said. However, there are many more interesting and challenging targets to perform protein engineering in the real world, for which developing accurate predictors is challenging due to the scarcity of data or the difficulty to model the proteins, as shown in ProteinGym. Therefore, I'm really concerned about whether the method still works under these realistic and challenging settings.
> >
> > That said, please note that the reviewer's critique is towards the evaluation protocol of the specific task, not to this specific paper. As I mentioned in the review - "I'm actively working on the protein domains, but am not familiar with the specific task", I'm open to hear more thoughts about how to make the evaluation more robust and make real usage of the methods in this domain.

---

> > > ### Author Response · Authors · 2025-04-03
> > >
> > > We want to thank the reviewer for clarifying their response and for raising the concern about the evaluation protocol in the context of protein fitness optimization.
> > >
> > > Since our focus was on exploring a method that is novel within this context, we believe it is important to first rely on established benchmarks to compare our approach with competing methods. Our goal in this work is to establish a strong foundation using existing benchmarks before extending the approach to more complex or under-explored proteins. The fact that VLGPO achieves robust performance even on the more difficult GFP (hard) task with only few sequences - where competing methods often fail to generate mutants with better fitness - is a strong indicator that it is worth exploring further. This highlights that the idea of using classifier guidance in a continuous latent space is very promising.
> > >
> > > We acknowledge the sentiment and the point raised by the reviewer and we agree that there is a need to extend the evaluation to other, potentially more challenging or diverse proteins from FLIP or ProteinGym. We also believe that both in-silico evaluation such as the work in [1] assessing the utility of computational filters and using metrics like folding confidence and structural stability (to name a few) could offer a more complete perspective beyond fitness evaluation. Additionally, we think that future research could adopt iterative optimization protocols that better simulate realistic settings and incorporate structural or experimental validation.
> > >
> > > However, we believe that developing and implementing such expansions is essential, but goes beyond the scope of our current work, as it constitutes a significant research effort on its own. We hope this clarifies our reasoning and demonstrates the potential of our approach, especially as a step toward more realistic protein engineering scenarios.
> > >
> > > [1] Johnson, Sean R., et al. "Computational scoring and experimental evaluation of enzymes generated by neural networks." *Nature biotechnology* (2024): 1-10.

---

### Official Review · Reviewer_jm6F · 2025-03-13

**Overall Recommendation:** 4

**Summary:**

The paper proposes Variational Latent Generative Protein Optimization (VLGPO), a method for protein sequence optimization by training a VAE combined with a learned flow matching prior over mutations. A fitness predictor is used for guidance, and the method is evaluated on commonly used database lookups including the AAV and GFP datasets, as well as diversity and novelty metrics. Hyperparameter optimization is identified as a challenge for the approach, which is sensitive to these choices.

**Claims And Evidence:**

Yes

**Essential References Not Discussed:**

The authors may be interested in this very recent in silico benchmark, which introduces synthetic test functions that can be used for benchmarking black-box optimization methods like VLGPO.

Stanton, S., Alberstein, R., Frey, N., Watkins, A., & Cho, K. (2024). Closed-form test functions for biophysical sequence optimization algorithms. arXiv preprint arXiv:2407.00236.

**Experimental Designs Or Analyses:**

Yes

**Methods And Evaluation Criteria:**

Yes

**Other Comments Or Suggestions:**

See above

**Other Strengths And Weaknesses:**

The paper is clearly written, addresses an important problem in biology, contains relevant benchmarks and baselines, and introduces a novel approach.

The biggest weakness is the reliance on GFP and AAV datasets as the benchmarks, which is a general limitation that affects the field. Have the authors considered other FLIP tasks, or other related tasks?

**Questions For Authors:**

See above

**Relation To Broader Scientific Literature:**

The paper is well-situated in terms of the broader literature, and detailed comparisons are made to prior work

**Theoretical Claims:**

N/A

---

> ### Author Rebuttal · Authors · 2025-03-27
>
> We sincerely thank the reviewer for their careful evaluation of our manuscript and for the positive assessment of our work. Below, we address each point individually:
>
> - **The authors may be interested in this very recent in silico benchmark, which introduces synthetic test functions that can be used for benchmarking black-box optimization methods like VLGPO.**
>
> Thank you for directing us toward this interesting recent work. We will examine it in more detail and consider its suitability for future applications, as synthetic test functions appear very promising for further assessing the generated sequences.
>
> - **The biggest weakness is the reliance on GFP and AAV datasets as the benchmarks, which is a general limitation that affects the field. Have the authors considered other FLIP tasks, or other related tasks?**
>
> This is indeed an important point. While our method could, in principle, extend to other FLIP or ProteinGym tasks, we chose the GFP and AAV datasets as they currently are well-established benchmarks to allow for fair comparisons against our proposed approach. The main focus here was to investigate performance in limited initial datasets, reflecting common practical scenarios.
>
> Nevertheless, exploring additional tasks beyond GFP and AAV is an exciting avenue for future research, and we fully agree on the importance of designing and adopting broader benchmarks in this context.

---

> > ### Comment · Reviewer_jm6F · 2025-04-03
> >
> > I appreciate the authors' response and their reply that extending benchmarking beyond the considered tasks is outside the scope of the current work. I will raise my score and recommend acceptance.

---

> > > ### Author Response · Authors · 2025-04-04
> > >
> > > We thank the reviewer and appreciate the raised score; we will revise the manuscript to incorporate all the initial feedback.

---

### Official Review · Reviewer_TQyg · 2025-03-14

**Overall Recommendation:** 3

**Summary:**

This paper presents a novel protein fitness optimization model called Variational Latent Generative Protein Optimization (VLGPO). VLGPO uses flow-matching to perform fitness optimization in the continuous latent space of the generative model, allowing efficient exploration of the fitness landscape. Guided by fitness predictors, VLGPO can effectively optimize high-fitness proteins.

**Claims And Evidence:**

The author demonstrated the performance of VLGPO on two protein datasets, GFP and AAV, with medium and hard difficulty. VLGPO showed superior fitness optimization ability compared to previous methods.

**Essential References Not Discussed:**

NA

**Experimental Designs Or Analyses:**

The author follows the experimental design in [Kirjner et al., 24], demonstrating that VLGPO can optimize sequences for both high fitness and high diversity and novelty. In Tables 2 and 3, the variability in diversity and novelty among different methods is significantly greater in GFP than in AAV, which is another observation worth discussing.

**Methods And Evaluation Criteria:**

The paper clearly presents the algorithmic and training details of flow-matching and classifier guidance. However, since the evaluation of the methods relies on the predictor $g_{\phi}$, it would be helpful to provide details on the accuracy of $g_{\phi}$ as in-silico oracle.

**Other Comments Or Suggestions:**

NA

**Other Strengths And Weaknesses:**

It would be helpful if the author could also provide more analysis and statistics, such as diversity at different fitness, of the GFP and AAV dataset.

**Questions For Authors:**

See above.

**Relation To Broader Scientific Literature:**

Previous works mainly focus on directly optimizing protein sequences. This paper optimizes protein sequence in a continuous latent space by using flow matching, providing novel insights into exploring the protein fitness landscape.

**Theoretical Claims:**

NA

---

> ### Author Rebuttal · Authors · 2025-03-27
>
> We would like to thank the reviewer for their valuable comments and feedback. Below, we address each point individually:
>
> - **Since the evaluation of the methods relies on the predictor, it would be helpful to provide details on the accuracy of $g_{\phi}$  as in-silico oracle.**
>
> Thank you for highlighting this important aspect. We reused the oracle and predictor from [1] entirely out-of-the-box, without any modifications, aiming to present a method in which alternative predictors or oracles could easily be used as drop-in replacements. We will comment on this in more detail in the revised version.
>
> However, since [1] does not explicitly report the final performance of the in-silico oracle, we have now computed the Mean Squared Error (MSE) on a subset of 512 randomly selected samples from the ground truth dataset $\mathcal{S}^*$. The oracle’s predictions closely follow the target fitness values, resulting in MSE values of 0.012240 for GFP and 0.002758 for AAV.
>
> - **In Tables 2 and 3, the variability in diversity and novelty among different methods is significantly greater in GFP than in AAV, which is another observation worth discussing.**
>
> We appreciate the reviewer’s insightful observation. This aligns well with our findings, indicating that the GFP setting indeed appears more challenging compared to AAV. This discrepancy may arise due to GFP sequences being longer and thus representing sparser and higher-dimensional search spaces. We will discuss this observation in the revised version of the manuscript.
>
> - **It would be helpful if the author could also provide more analysis and statistics, such as diversity at different fitness, of the GFP and AAV dataset.**
>
> Thank you for this valuable suggestion. The diversity within the top-performing (99th percentile) sequences of the entire dataset $\mathcal{S}^*$ is 4.73 for GFP and 5.23 for AAV, as briefly discussed in Section 4.3. In contrast, the diversity of sequences in the four considered tasks is notably higher: for GFP (medium) it is 14.5, for GFP (hard) 16.3, for AAV (medium) 15.9, and for AAV (hard) 18.4. The training datasets naturally contain more diverse sequences, as it includes those with lower fitness levels. In contrast, the top-performing sequences exhibit lower diversity, aligning closely with the diversity observed in Tables 2 and 3. We will highlight this observation in the revised manuscript.
>
>  [1] Kirjner, Andrew, et al. "Improving protein optimization with smoothed fitness landscapes." *ICLR,* 2024.

---

### Official Review · Reviewer_3GwA · 2025-03-17

**Overall Recommendation:** 4

**Summary:**

This paper proposes a new in-silico method for generating novel high-fitness protein sequences. It first embeds sequences in a lower-dimensional space via a VAE, then fits a generative model to the embeddings by flow-matching. The sampling is guided by a pre-trained fitness predictor and manifold-constrained gradients. Empirical results show improved sampling of high-fitness variants, even in data-scarce settings.

**Claims And Evidence:**

* This paper argues that, for protein fitness optimisation, embedding sequences in a continuous latent space is more effective than token-based sequence representation. This claim is partially supported by the comparison with GWG, GGS, and gg-dWJS that rely on one-hot encoding.
* This paper argues that using a fitness predictor is an effective approach to guide the optimisation towards high-fitness regions, particularly in limited data regimes. This claim is supported by the experimental study, notably with the comparison with a conditional diffusion model trained from scratch (on the labeled sequence-fitness pairs) without the fitness predictor guidance. The supplementary material reports the different evaluation metrics calculated on the samples obtained from unconditional diffusion model, for reference.

**Essential References Not Discussed:**

The related work is comprehensive.

**Experimental Designs Or Analyses:**

The experimental design and analyses are solid:
* Results are averaged over five seeds.
* A fair comparison with other baselines is achieved through grid search over hyperparameters and using the same networks when possible.
* The ablation study (comparing guidance with and without manifold-constrained gradients, unconditional, and conditioning from scratch) is conducted for both cases, with and without graph-based smoothing.

**Methods And Evaluation Criteria:**

* The method is sound, well-motivated and each component is clearly described in Section 3.
* The evaluation is robust. Employing two protein optimisation benchmarks of varying complexity, 9 baseline approaches, and 3 different metrics, it clearly demonstrates the advantages of the proposed approach.
* However, as the authors note among the potential limitations, the assessment of the fitness relies on a neural fitness predictor which is probably less reliable than wet-lab validation.

**Other Comments Or Suggestions:**

* Can you clarify the connections between the proposed approach and variational inference?

**Other Strengths And Weaknesses:**

**Strengths**
- The clarity of the paper.
- The robust experimental design.
- The novel combination of ideas applied to the real-world use case of protein fitness optimisation.

**Weaknesses**
- The use of the fitness predictor may overestimate the performance gap with some baseline approaches.

**Questions For Authors:**

* Can you comment on the poor performance of GFN-AL?
* Can you contrast your approach with classifier-free guidance?

**Relation To Broader Scientific Literature:**

The method is a combination of ideas from the latest generative modelling literature (flow-matching in VAE latent space and guidance with manifold constrained gradient) applied to the real-world use case of protein fitness optimisation.

**Theoretical Claims:**

N/A

---

> ### Author Rebuttal · Authors · 2025-03-27
>
> We would like to thank the reviewer for the constructive feedback and the insightful comments, which help to improve the quality of the manuscript.
>
> We fully agree with the reviewer that assessing fitness with a computational model is less reliable than direct wet-lab validation. Experimental validation is the final goal, in-silico evaluation is an important step along the way. We also appreciate the comment regarding the potential overestimation of the performance gap compared to baseline methods. This is indeed a challenge for the community, but hard to quantify precisely.
>
> Below, we address the reviewer’s more specific questions directly:
>
> - **Can you clarify the connections between the proposed approach and variational inference?**
>
> Thank you for this point. VLGPO uses variational inference through the initial embedding step to obtain continuous latent embeddings of the discrete protein sequence variants - the autoencoder is trained with variational inference. However, our approach extends this by introducing additional generative modeling components,  namely flow matching and classifier-guided sampling, to further refine and control the generation process. We will clarify this connection explicitly in the revised manuscript.
>
> - **Can you comment on the poor performance of GFN-AL?**
>
> To ensure a fair evaluation, the results for all methods are taken from a comprehensive comparison done in recent published work [1]. Hence we can only speculate about the poor performance. GFN-AL might struggle in limited-data scenarios due to sparse reward signals from the few observed mutants, which could potentially limit effective exploration.
>
> - **Can you contrast your approach with classifier-free guidance?**
>
> Thank you for this question — we find it very interesting, as it highlights the advantages of using a separate classifier for guidance, which we have addressed in Fig. 3. There, we compare our method to a conditional model that directly learns the posterior $p(x|y)$. In that setup we additionally tested classifier free guidance, but it did not make a noticeable difference. As shown in both the figure and the ablation tables, classifier guidance allows for  more effective steering of the generation process: in challenging scenarios like GFP (hard), classifier-free guidance struggles to produce higher-fitness sequences, while classifier guidance successfully targets high-reward regions.
>
>  [1] Kirjner, Andrew, et al. "Improving protein optimization with smoothed fitness landscapes." *ICLR,* 2024.

---

### Decision · Program_Chairs · 2025-05-01

**Decision:**

Accept (poster)

**Comment:**

This paper proposes to optimize the fitness of proteins by first compressing the protein sequences into latent continuous space and then using flow matching models to sample in the latent space guided by a pretrained protein fitness predictor. Experimental results prove the competitiveness of the proposed approach on two protein benchmarks.

The reviewers in general appreciate the novelty of the proposed approach and its significance over existing approaches. A reviewer raises big concerns over the evaluation set up, which the authors should address in revised version. The AC tends to vote for acceptance.